# Semaglutide Improves Lipid Subfraction Profiles in Type 2 Diabetes: Insights from a One-Year Follow-Up Study

**DOI:** 10.3390/ijms26135951

**Published:** 2025-06-20

**Authors:** László Imre Tóth, Adrienn Harsányi, Sára Csiha, Ágnes Molnár, Hajnalka Lőrincz, Attila Csaba Nagy, György Paragh, Mariann Harangi, Ferenc Sztanek

**Affiliations:** 1Division of Metabolism, Department of Internal Medicine, Faculty of Medicine, University of Debrecen, H-4032 Debrecen, Hungary; toth.laszlo@med.unideb.hu (L.I.T.); lharsanyi18@gmail.com (A.H.); molnar.antalne@med.unideb.hu (Á.M.); lorincz_hajnalka@belklinika.com (H.L.); paragh@belklinika.com (G.P.); harangi@belklinika.com (M.H.); 2Doctoral School of Health Sciences, University of Debrecen, H-4032 Debrecen, Hungary; csiha.sara@med.unideb.hu; 3Department of Clinical Basics, Faculty of Pharmacy, University of Debrecen, H-4032 Debrecen, Hungary; 4Department of Health Informatics, Faculty of Health Sciences, University of Debrecen, H-4032 Debrecen, Hungary; nagy.attila@etk.unideb.hu; 5Institute of Health Studies, Faculty of Health Sciences, University of Debrecen, H-4032 Debrecen, Hungary; 6ELKH-UD Vascular Pathophysiology Research Group 11003, University of Debrecen, H-4032 Debrecen, Hungary

**Keywords:** type 2 diabetes mellitus, semaglutide, sitagliptin, lipoprotein subfractions, GLP-1 receptor agonist, cardiometabolic risk

## Abstract

Recent studies have demonstrated the efficacy of glucagon-like peptide-1 receptor agonists and dipeptidyl peptidase-4 inhibitors in enhancing glycemic control, regulating body weight, and modulating lipid metabolism. However, their effects on lipoprotein subfractions have not been clarified. The objective of this 52-week, single-center, randomized trial was to compare the effects of subcutaneous semaglutide administered once weekly and oral sitagliptin administered once daily on anthropometric measurements and lipoprotein subfractions measured by Lipoprint gelelectrophoresis in patients with type 2 diabetes mellitus (T2DM). A total of 34 obese individuals with T2DM were enrolled in the study and randomly assigned to receive semaglutide (*n* = 18) or sitagliptin (*n* = 16). Thirty-one age- and body weight-matched non-diabetic obese individuals served as controls. Semaglutide treatment resulted in significant reductions in body mass index (BMI), waist circumference, and Hb_A1c_, along with improvements in lipid parameters, including reductions in LDL cholesterol and non-HDL cholesterol levels, and redistribution of LDL and HDL subfractions toward a less atherogenic profile. Conversely, sitagliptin elicited modest glycemic improvements without substantial alterations in lipid composition. Multivariate regression analysis demonstrated that fluctuations in lipoprotein subfractions were not influenced by changes in BMI or HbA1c. These results support the pleiotropic metabolic benefits of semaglutide and its potential role in managing the cardiometabolic risk of T2DM.

## 1. Introduction

Cardiovascular disease remains the leading cause of morbidity and mortality in patients with type 2 diabetes mellitus (T2DM), with dyslipidaemia being a critical risk factor alongside hyperglycaemia. Given the strong association between lipid abnormalities and cardiovascular complications, correction of lipid abnormalities is a key therapeutic target in the management of patients with diabetes [1].

Obesity- and diabetes-related dyslipidaemia is a well-documented metabolic complication in obese individuals, characterised by elevated concentrations of triglyceride-rich lipoproteins, i.e., very low-density lipoprotein (VLDL), intermediate density lipoprotein, small-dense low-density lipoprotein (sdLDL), as well as a reduction in mean low-density lipoprotein (LDL) particle size [2]. In addition, we observed a shift towards smaller high-density lipoprotein (HDL) subfractions in non-diabetic obese patients compared to lean individuals, potentially contributing to an altered lipid profile that exacerbates cardiovascular risk [3]. Emerging evidence from the large, prospective, population-based PREVEND study suggests that distinct patterns of association exist between obesity, triglyceride levels, LDL particle characteristics, and the risk of developing T2DM. Specifically, elevated levels of triglyceride-rich lipoproteins are associated with an increased risk of T2DM. In contrast, larger LDL particle size and greater LDL particle diameter appear to be associated with a decreased risk of diabetes, potentially providing protection against the development of the disease [4]. These findings underscore the complex interplay between lipoprotein metabolism and metabolic disorders and reinforce the need for targeted lipid-lowering strategies in individuals at high cardiometabolic risk.

Glucagon-like peptide-1 (GLP-1), an incretin hormone produced in the gut, plays a critical role in carbohydrate homeostasis by stimulating insulin secretion from pancreatic β-cells in response to nutrient intake. In addition, GLP-1 delays gastric emptying and promotes weight loss by suppressing appetite, a mechanism that is particularly beneficial in patients with T2DM who have a reduced natural incretin effect [5,6]. Administration of glucagon-like peptide-1 receptor agonists (GLP-1RAs) enhances this effect, improving glycaemic control while exerting beneficial effects on lipid metabolism. GLP-1RAs have been shown to reduce levels of low-density lipoprotein cholesterol, very-low-density lipoprotein cholesterol, total cholesterol and triglycerides, although no significant increase in HDL cholesterol has been observed [7,8]. However, experimental studies suggest that GLP-1RAs may ameliorate HDL functionality and increase the concentration of smaller HDL particles, contributing to the atheroprotective properties of HDL [9].

Among the GLP-1RAs, semaglutide has emerged as a particularly effective agent in the treatment of T2DM and is currently being investigated for its potential applications in weight management. Clinical trials have consistently shown that semaglutide-induced weight loss improves glycaemic control in patients with obesity and T2DM, while also reducing cardiovascular risk in high-risk populations [10,11]. In addition, liraglutide, another GLP-1RA, has been shown to reduce apolipoprotein B100 (apoB) levels and increase LDL particle size, thereby reducing the concentration of sdLDL particles, which are highly atherogenic [8]. Taken together, these findings highlight the potential of semaglutide not only as a glucose-lowering agent, but also as a modulator of lipid metabolism and cardiovascular risk, reinforcing its role in the comprehensive management of T2DM and associated cardiometabolic disorders [12].

Dipeptidyl peptidase-4 (DPP-4) inhibitors have shown favorable pleiotropic effects on atherosclerosis to improve endothelial cell function, which is essential for maintaining vascular integrity and preventing atherosclerotic progression [13]. DPP-4 inhibitors modulate lipid metabolism by reducing levels of triglyceride-rich lipoproteins, triglycerides, and apoB [14,15]. Although, no significant increase in HDL cholesterol levels has been observed, experimental studies suggest that DPP-4 inhibitors may improve HDL function and enhance its atheroprotective properties [16]. In addition to lipid regulation, DPP-4 inhibitors exert significant anti-inflammatory and antioxidant effects, thereby attenuating key contributors to atherosclerosis [16]. The above-mentioned mechanisms underscore the potential of DPP-4 inhibitors to improve disease management and patient outcomes in individuals with T2DM.

To date, the effect of semaglutide and sitagliptin on LDL and HDL subfractions has not been investigated. Therefore, the aim of this study was to evaluate and compare the long-term effect of the GLP-1 RA semaglutide and the DPP-4 inhibitor sitagliptin on body weight and lipid metabolism in patients with T2DM after a 26- and 52-week follow-up. Specifically, this study assesses changes in lipid parameters and the distribution of LDL and HDL subfractions, with a particular focus on the effect of subcutaneous administration of semaglutide and oral sitagliptin on the lipid profile. We hypothesized that the effect of semaglutide on LDL and HDL subfraction alteration may be superior compared to the effect of sitagliptin.

## 2. Results

### 2.1. Baseline Clinical and Laboratory Parameters in Enrolled Subjects

The clinical and laboratory characteristics of T2DM patients before treatment with semaglutide or sitagliptin and control subjects are summarised in Table 1.

There were no significant differences in age, body mass index (BMI), waist circumference, creatinine, liver enzymes, or lipid parameters between the two groups of patients before treatment and between controls. Fasting blood glucose, fructosamine, Hb_A1c_ and C-peptide levels were significantly higher in both diabetic groups compared to controls (*p* < 0.05). There were no significant differences in baseline levels of LDL and HDL subfractions between the semaglutide and sitagliptin groups.

### 2.2. Changes in Clinical and Laboratory Parameters After a 52-Week Semaglutide Treatment in Type 2 Diabetic Patients

The changes in anthropometric and laboratory parameters after 26 and 52 weeks of semaglutide treatment are summarized in Table 2.

A marked reduction in BMI was observed over time, from 37.96 ± 10.64 kg/m^2^ at baseline to 35.28 ± 9.43 kg/m^2^ at week 26 (*p* < 0.05) (−7.1%) and further decreased to 34.88 ± 10.22 kg/m^2^ at week 52. This resulted in a mean weight reduction of 8.1% by the end of the study. Similarly, waist circumference decreased significantly from 126.4 ± 21.8 cm at baseline to 119.7 ± 21.6 cm at week 26 (*p* < 0.05) and 115.5 ± 20.9 cm at week 52 (*p* < 0.05) (−5.3% and −8.6%, respectively). Fasting glucose and fructosamine decreased steadily over time (both *p* < 0.05) and Hb_A1c_ also decreased significantly, from a baseline of 8.08 ± 1.65% to 6.86 ± 1.12% at week 26 (*p* < 0.05) and then decreased further to 6.57 ± 0.95% (*p* < 0.05), with an average Hb_A1c_ reduction of 1.5% at week 52. Triglyceride and total cholesterol levels showed a non-significant decreasing trend, while LDL and non-HDL cholesterol levels showed a significant decrease at week 52 (*p* < 0.05). In addition, HDL cholesterol was significantly elevated at week 26 and remained elevated at week 52 (both *p* < 0.05). Both large and small LDL subfractions showed significant decreases at week 26 (*p* < 0.05), which remained significant at week 52 (*p* < 0.05). Analysis of individual LDL subfractions, particularly LDL-2 and LDL-3 subfractions showed a significant reduction with semaglutide treatment at week 52 of the study (both *p* < 0.05, Figure 1). Levels of the large HDL subfractions increased significantly after week 26 (*p* < 0.05) but returned to baseline levels by week 52. Intermediate HDL subfractions also showed a significant increase at 26 and 52 weeks (*p* < 0.05); while small HDL subfractions remained unchanged. Among the individual HDL subfractions, the HDL-1 subfraction increased significantly (*p* < 0.05); while the HDL-6 and HDL-10 subfractions decreased significantly (*p* < 0.05 and *p* < 0.01, respectively) after semaglutide treatment at week 52 (Figure 2). Apolipoprotein A1 (apoA1) and apoB levels did not change significantly over time. The inflammatory marker hsCRP showed a slight decrease, but this was not statistically significant. Renal function parameters, including creatinine levels and eGFR, remained stable over the study period.

### 2.3. Changes in Clinical and Laboratory Parameters After a 52-Week Sitagliptin Treatment in Type 2 Diabetic Patients

Changes in anthropometric and laboratory parameters after 26 and 52 weeks of sitagliptin treatment are summarized in Table 3.

A modest decrease in BMI was observed over time (*p* < 0.05), while waist circumference showed a non-significant decrease over the 52-week period. Plasma glucose, fructosamine, and HbA1c decreased significantly at week 26 and week 52, respectively (Table 3). No significant changes were observed in insulin, C-peptide, hsCRP, and lipid levels. In addition, lipoprotein subfractions, liver enzymes, and renal function parameters remained stable during the follow-up as well (Table 3; Figure 1 and Figure 2). Interestingly, apoA1 levels decreased significantly from baseline to 26 weeks (*p* < 0.05) and further decreased to 52 weeks (*p* < 0.05). ApoB levels did not change significantly over time.

### 2.4. Multivariate Linear Regression Model to Analyze Changes in Lipid Subfractions Induced by Semaglutide Treatment

To analyze the changes in lipoprotein subfractions after semaglutide treatment, a multivariate linear regression model was constructed and is summarised in Appendix A. The analysis was performed to assess the association between changes in LDL and HDL subfractions after semaglutide treatment and improvement in BMI, adjusting for age, sex and changes in HbA1c levels (Table 4). The analysis showed no statistically significant associations between changes in LDL and HDL subfractions and BMI, age, sex or changes in HbA1c levels (*p* > 0.05 for all variables).

## 3. Discussion

This study is the first to demonstrate the effect of semaglutide treatment on LDL and HDL subfractions in obese T2DM patients in a clinical setting, supporting significant metabolic benefits in addition to effective glycaemic and weight-lowering effects. The atherogenic lipid profile improved, LDL and non-HDL cholesterol levels decreased, HDL cholesterol increased, and the LDL and HDL subfractions changed favorably as well. Sitagliptin treatment showed more modest reductions in body weight and Hb_A1c_, while lipid parameters and inflammatory markers changed less, and the stability of the effects was moderate.

Semaglutide has emerged as a highly effective glucose-lowering agent in the treatment of both obesity and T2DM, with significant weight loss and metabolic benefits. Several studies have shown that semaglutide improves glycaemic control in T2DM and promotes significant weight loss. The SUSTAIN-6 trial, a multicenter, randomized, placebo-controlled study, demonstrated that once-weekly subcutaneous administration of 1 mg semaglutide in patients with T2DM resulted in a significant 1.5% reduction in HbA1c, an average weight loss of approximately 4.5 kg, and a 26% decrease in major adverse cardiovascular events over 104 weeks [17]. In this study, we applied the findings from SUSTAIN-6 to assess the effects of semaglutide on anthropometric, glycemic, and lipid parameters over 52 weeks in obese patients with T2DM, observing consistent reductions in HbA1c and body weight, along with favorable changes in lipoprotein subfractions and lipid profiles, thereby corroborating and extending the evidence from large randomized clinical trials. The STEP 1 trial demonstrated a mean weight loss of 14.9% in obese, non-diabetic subjects treated with 2.4 mg semaglutide per week [18]. Similarly, the results of the STEP 2 trial supported the concept that semaglutide-induced weight loss is significant and durable in obese individuals with T2DM, demonstrating a 9.6% weight loss at week 68 with subcutaneous semaglutide 2.4 mg once weekly [19]. In this study, the mean weight loss was 8.1% and the mean reduction in HbA1c was 1.5% with 1 mg semaglutide once weekly at 52 weeks in obese T2DM subjects, which are consistent with the results of large, multicentre randomized clinical trials.

Semaglutide exerts its effects by promoting glucose-dependent insulin secretion and inhibiting glucagon release, while its weight-loss properties are attributed to delayed gastric emptying and modulation of central nervous system appetite regulation. Consequently, these mechanisms improve glycaemic control and reduce overall energy intake [19]. In addition to its efficacy in the treatment of obesity, semaglutide has been demonstrated to offer several cardiovascular benefits, including an improvement in lipid profiles, a reduction in the progression of atherosclerotic cardiovascular disease, and lowered blood pressure, as evidenced by long-term studies [17]. The magnitude of weight reduction is positively correlated with the dosage, with higher doses demonstrating a greater potential for substantial outcomes. It is imperative to emphasise the necessity of long-term adherence to treatment, as the sustained use of the medication is crucial in determining its tolerability and the sustainability of weight loss outcomes. This assertion is further substantiated by the findings of the STEP 4 trial, which demonstrated that the discontinuation of treatment resulted in substantial weight regain [20]. This study emphasises the critical role of sustained semaglutide therapy in achieving and maintaining long-term metabolic health improvements.

This study demonstrated a significant reduction of LDL and non-HDL cholesterol levels, while total cholesterol levels exhibited no substantial downward trend following 52 weeks of subcutaneous semaglutide treatment. In addition, we observed substantial and persistent reductions in both large and small LDL subfractions. Subsequent analysis of individual LDL subfractions revealed significant decreases in LDL-2 and LDL-3 after semaglutide treatment. These outcomes are consistent with prior studies, demonstrating that semaglutide significantly reduces LDL and non-HDL cholesterol levels in obese patients with T2DM, both in clinical and real-world settings [21,22]. Furthermore, semaglutide has been associated with long-term and sustained improvements in lipid abnormalities, including significant reductions in LDL and non-HDL cholesterol [23]. The GLP-1 RA liraglutide has been demonstrated to exert a positive influence on LDL cholesterol and associated cardiovascular risk factors by diminishing oxidized LDL-induced endothelial dysfunction, enhancing dyslipidaemia, and augmenting the catabolism of apoB-containing lipoproteins [8,24]. Furthermore, a preponderance of evidence from in vitro and animal studies has demonstrated that both liraglutide and semaglutide offer protection against oxidized LDL-induced endothelial inflammation and dysfunction. This protective effect is achieved through the activation of the AMPK/ABCA1 and Krüppel-like Factor 2 pathways, thereby slowing the progression of atherosclerosis [25,26]. The available animal models indicate that the cardiovascular benefits of semaglutide extend beyond the modulation of the atherogenic lipid profile. This is evidenced by the reduction in atherosclerotic plaque formation in ApoE-/- and LDLr-/- mice, which occurs independently of weight loss and cholesterol reduction [27]. A substantial body of research has repeatedly demonstrated that sdLDL particles possess a markedly elevated atherogenic potential in comparison to the large LDL subfraction. This heightened potential is attributed to several key factors, including their enhanced capacity to infiltrate the arterial intima, a prolonged plasma half-life, and an augmented susceptibility to oxidative modifications [28]. These oxidative changes, which are facilitated by enzyme activities such as myeloperoxidase and lipoprotein-associated phospholipase A2, promote endothelial dysfunction and local vascular inflammation, thereby also enhancing foam cell formation and atherosclerotic plaque progression [29,30]. In addition, increased hepatic lipase activity, characterized by a shift towards a denser LDL phenotype, has been demonstrated to augment cardiovascular risk by increasing the circulatory burden of sdLDL particles. This underscores the importance of treating LDL cholesterol as a risk factor and its targeted therapeutic target in determining the LDL subfraction [31]. The findings of this study demonstrate that semaglutide treatment leads to a substantial decrease in LDL- and non-HDL cholesterol levels, with a particular reduction in LDL-2 and LDL-3 subfractions. Furthermore, the treatment has been demonstrated to improve the management of lipid abnormalities and may offer cardiovascular protection.

This study demonstrated a significant increase in serum HDL cholesterol levels after 26 weeks of semaglutide treatment, which remained persistently elevated at week 52. In particular, the large HDL subfraction increased significantly after week 26 before returning to baseline at week 52, whereas the intermediate HDL subfractions showed a sustained increase at both time points and the small HDL subfractions remained unchanged; among the individual HDL subfractions, HDL-1 levels increased significantly, whereas HDL-6 and HDL-10 levels decreased significantly at the end of the study. Previously, a pilot study in Japanese patients with T2DM showed significant reductions in total cholesterol, LDL cholesterol, and non-HDL cholesterol during the first month of daily subcutaneous liraglutide treatment, while HDL cholesterol levels remained unchanged after three months [32]. However, our results are consistent with recent studies showing that subcutaneous semaglutide, which is more potent than liraglutide, significantly increases HDL cholesterol levels when administered weekly [23]. In addition, real-world data show that semaglutide also significantly reduces proatherogenic non-HDL cholesterol levels after 3, 6, and 12 months of treatment [22]. In addition, a pooled post hoc analysis of the STEP 1 and 4 studies in overweight or obese subjects without diabetes further supports the ability of semaglutide to improve cardiometabolic risk factors, including beneficial modification of the abnormal lipid profile [33]. Similarly, preclinical studies show that the GLP-1 receptor liraglutide improves glucose and lipid metabolism by lowering blood glucose, serum total cholesterol, triglyceride, and LDL cholesterol levels, reducing hepatic lipid accumulation, increasing macrophage cholesterol efflux, and upregulating ABCA1 expression, which is critical for HDL synthesis, through extracellular signal-regulated kinase 1/2 pathway activation, as demonstrated in db/db mice on a high-fat diet and in HepG2 cells under high-glucose conditions [26]. In addition, liraglutide has been shown to stabilize apoB levels during weight maintenance and to modify HDL distribution [34,35]. Preliminary research indicates that the size and composition of HDL subfractions significantly influence their functional properties. Large HDL particles have been shown to exert a pronounced atheroprotective effect by enhancing reverse cholesterol transport, anti-inflammatory effects, and improving endothelial function [36]. Conversely, smaller HDL subfractions, particularly in the presence of oxidative stress or metabolic disorders, may undergo compositional changes that impair antioxidant capacity and promote a pro-inflammatory phenotype, thereby contributing to atherogenesis [37]. These mechanistic insights imply that the net cardiovascular effect of HDL is not solely determined by its plasma concentration, but rather by the dynamic transformation and context-dependent functionality of subfractions. These subfractions are modulated by key enzymes such as lecithin–cholesterol acyltransferase, cholesteryl ester transfer protein, human paraoxonase-1, and endothelial lipase [38]. Taken together, these results confirm the role of semaglutide in improving HDL cholesterol levels and broader lipid metabolism, possibly through increased cholesterol efflux and modulation of lipid metabolism, suggesting potential cardiovascular benefits in diverse patient populations.

In this study, we observed that sitagliptin treatment did not result in significant changes in total, LDL, HDL, or non-HDL cholesterol levels over a 52-week period. In addition, no changes were observed in large or small LDL subfractions. Although large HDL subfractions showed a slight, non-significant increase, medium and small HDL subfractions remained unchanged. ApoA1 levels decreased significantly from baseline at 26 weeks and continued to improve at 52 weeks, while apoB levels remained unchanged. These results are in contrast with the results of several meta-analyses and clinical trials that have shown improvements in lipid profiles with sitagliptin therapy. For instance, a meta-analysis of 11 randomized clinical trials reported significant improvements in serum triglyceride and HDL cholesterol levels, despite no change in LDL cholesterol or total cholesterol [39]. Conversely, a systematic review of 14 randomized clinical trials showed that sitagliptin significantly reduced total and LDL cholesterol, while triglyceride and HDL cholesterol levels remained largely unchanged [40]. Further analysis in T2DM patients showed significant reductions in total cholesterol and non-HDL cholesterol, particularly in individuals with elevated baseline triglyceride levels [41]. The observed differences may be due to differences in study designs, populations, and treatment duration, underscoring the need for further research into the lipid-modulating effects of sitagliptin.

This study has several strengths, including a randomized design, integrating patient groups with clearly defined inclusion and exclusion criteria alongside a matched non-diabetic control group, which allowed for comprehensive comparative analysis. In addition, the longitudinal 52-week follow-up with repeated measurements strengthens the temporal validity of the findings. The rigorous methodological approach is demonstrated by standard laboratory procedures, such as detailed lipoprotein subfraction analysis by validated electrophoresis, and detailed statistical analyses further enhance the reliability and interpretability of the results. Despite its methodological rigor, this study has several limitations that require consideration. The single-center design, limited sample size, and lack of racial diversity restrict the generalizability of the results, highlighting the need for further studies in more diverse populations. This study’s narrow inclusion criteria, which focus on middle-aged, obese patients with long-standing T2DM, further restrict the external validity of the results. The fixed dosing regimen of sitagliptin and the brief titration period of semaglutide may also influence therapeutic outcomes, while the reliance on surrogate endpoints, such as lipid subfraction analyses, instead of direct clinical outcomes, reduces the ability to determine long-term efficacy and cardiovascular effects. Another limitation of this study is the absence of a detailed analysis of lipoprotein particle composition and functional properties. While standard lipid parameters and subfraction distributions were examined, incorporating advanced proteomic and lipidomic profiling of LDL and HDL particles could have provided deeper insight into qualitative differences between groups. Additionally, quantifying phospholipid and cholesterol ester content and conducting functional studies, such as assessing HDL-mediated cholesterol efflux capacity in macrophages, would have improved our mechanistic understanding of the observed metabolic changes. Finally, potential confounding variables, including lifestyle factors and physical activity, were not comprehensively controlled for, necessitating a cautious interpretation of the study’s conclusions.

## 4. Materials and Methods

### 4.1. Study Design and Participants

This study was a single-centre, 52-week, randomized trial designed to evaluate the efficacy of the subcutaneous GLP-1RA semaglutide added to metformin monotherapy compared with the combination of baseline therapy with the DPP-4 inhibitor sitagliptin in patients with T2DM. A total of 34 adults with T2DM and obesity were enrolled. After randomisation, 18 patients (12 men, 6 women; mean age: 57.9 ± 10.6 years; mean BMI: 37.96 ± 10.64 kg/m^2^; mean duration of T2DM: 8.4 ± 3.6 years) were started on once-weekly subcutaneous semaglutide and 16 patients (7 men, 9 women; mean age: 11.9 years; mean BMI: 31.26 ± 2.75 kg/m^2^; mean duration of T2DM: 9.6 ± 3.2 years) were started on once-daily oral sitagliptin. All diabetic participants were on metformin monotherapy, received statin therapy, and the majority were also treated with an ACE inhibitor, angiotensin receptor blocker, or beta-blocker therapy. In addition, 31 age- and gender-matched obese, non-diabetic subjects (10 men, 21 women; mean age: 55.2 ± 5.0 years; mean BMI: 37.89 ± 9.64 kg/m^2^) were included as controls who were free of medication. Non-diabetic, obese control subjects were included to distinguish metabolic alterations related to T2DM from those associated with obesity alone.

The inclusion criteria were age over 45 years, T2DM for at least 5 years, inadequate glycaemic control with baseline metformin monotherapy (Hb_A1c_: 6.5–10%), BMI > 25 kg/m^2^, and eGFR > 60 mL/min/1.73 m^2^. The main exclusion criteria were type 1 diabetes mellitus, previous treatment with a GLP-1RA or DPP-4 inhibitor, active smoking, alcohol consumption, pregnancy, breastfeeding, documented coronary or cerebrovascular event, clinically significant heart failure (New York Heart Association stage II–IV), acute pancreatitis, pancreatic or thyroid cancer, severe liver failure (Child Pugh stage B and C), chronic use of systemic glucocorticoids or immunosuppressive drugs, advanced diabetic retinopathy (proliferative stage), diabetic nephropathy or severe renal impairment due to other causes (eGFR < 60 mL/min/1.73 m^2^ and/or significant proteinuria), and currently active malignant disease (remission of previous cancer > 5 years). All participants were recruited by the staff of the Division of Metabolism, Department of Internal Medicine, Faculty of University of Debrecen, Hungary. All patients gave written informed consent to participate in the study. The study protocol was approved by the local and regional ethics committee (RKEB/IKEB: 4739/2017, date of approval: 20 February 2017). This study was conducted in accordance with the Declaration of Helsinki.

In patients receiving semaglutide, the target dose was reached at week 8 after a titration period (starting dose: 0.25 mg/week subcutaneously; increased to 0.5 mg/week at week 4; and 1 mg/week at week 8). Sitagliptin was administered at a fixed dose of 100 mg daily from the start of treatment. Blood samples were taken immediately before the start of treatment and at routine check-ups after 26 and 52 weeks. The study design flowchart of diabetic patients on semaglutide and sitagliptin treatment are depicted on Figure 3.

### 4.2. Blood Sampling

Venous blood samples were collected after an overnight fast and sera were prepared immediately. Routine laboratory analyses, including triglycerides, total cholesterol, LDL cholesterol, HDL cholesterol, creatinine, glucose, insulin, C-peptide, Hb_A1c_, and liver enzymes were performed from fresh sera using a Cobas c501 autoanalyzer (Roche Ltd., Mannheim, Germany) at the Department of Laboratory Medicine, Faculty of Medicine, University of Debrecen, Hungary. Sera for subsequent determinations were stored at −70 °C until analysis. Reagents were purchased from the same supplier and assays were performed according to the manufacturer’s recommendations.

### 4.3. Lipoprotein Subfraction Analyses

LDL and HDL lipoprotein subfractions were separated using electrophoresis on polyacrylamide gel with Lipoprint System (Quantimetrix Corporation, Redondo Beach, CA, USA), following previously described procedures [42]. Liposure Serum Lipoprotein Control used as an internal standard in each electrophoresis chamber was provided by the manufacturer as well (Quantimetrix Corporation, Redondo Beach, CA, USA). Lipoprotein subfraction bands were digitalized using an ArtixScan M1 digital scanner (Microtek International Inc., San Diego, CA, USA) and analyzed using Lipoware Image SXM v.1.82 Software (Quantimetrix Corporation, Redondo Beach, CA, USA).

We distributed the lipoprotein subfractions using the Lipoprint LDL test (Cat. No. 48-7002, Quantimetrix Corporation, Redondo Beach, CA, USA). In addition to the VLDL, intermediate density lipoprotein and HDL bands, up to seven LDL subfractions were separated. After electrophoresis, lipoprotein fractions (bands) were identified by their mobility (Rf) using very low-density lipoprotein (VLDL) as the starting reference point (Rf 0.0) and HDL as the ending reference point (Rf 1.0). The LDL subfractions were distributed between these two points, from LDL1 to LDL7 (Rf 0.32, 0.38, 0.45, 0.51, 0.56, 0.6, and 0.64, respectively). The LDL1 and LDL2 bands corresponded to large, buoyant LDL subclasses, while the LDL3 to LDL7 bands corresponded to small, dense LDL subclasses. The percentages of the area under the curve (AUC) were calculated using Lipoware computer software (Quantimetrix Corp.). The proportion of large LDL (large LDL %) was calculated as the sum of the percentages of LDL-1 and -2, while the proportion of small LDL (small-dense LDL %) was determined as the sum of LDL-3-7. Cholesterol concentrations of LDL subfractions were calculated by multiplying the relative AUC of subfractions by the total cholesterol concentration. Total LDL cholesterol was computed as the sum of cholesterol in intermediate density lipoprotein (MidA-C) and LDL subfractions (LDL-1-LDL-7), which correlates with directly determined LDL cholesterol [43].

A Lipoprint HDL test was used for the distribution of HDL subfractions (Cat. No. 48-9002, Quantimetrix Corporation, Redondo Beach, CA, USA). The stained HDL subfractions (bands) were identified by their mobility after electrophoresis. The LDL/VLDL band served as the initial reference point (Rf 0.0), while albumin served as the final reference point (Rf 1.0). Percentages of the AUC were calculated using the Lipoware computer software (Quantimetrix Corp.). Ten HDL subfractions were determined, i.e., large (HDL-1,-2,-3), intermediate (HDL-4,-5,-6,-7), and small (HDL-8,-9,-10) subfractions. Cholesterol contents of HDL subfractions were determined using the Lipoware Software based on the relative AUC of subfraction bands [42].

We visually depicted the electrophoresis images of the LDL and HDL subfractions of each patient in the Appendix A to illustrate the analytical process.

### 4.4. Statistical Methods

Statistical analyses were performed using Statistica 13.5.0.17 software (TIBCO Software Inc., Palo Alto, CA, USA). Graphs were made by GraphPad Prism 6.01 (GraphPad Prism Software Inc., San Diego, CA, USA). Normality of variables was checked using the Shapiro–Wilk test. Continuous variables were described using medians (interquartile ranges) or means (standard deviation). Group comparisons were conducted using Kruskal–Wallis H test (with Bonferroni correction) for continuous variables. Repeated measures ANOVA (with Bonferroni correction) was employed to analyze the statistical changes in continuous variables during the 26- and 52-week semaglutide or sitagliptin treatment compared to baseline, respectively. A multivariate linear regression model was constructed to analyze changes in lipid subfractions induced by semaglutide treatment. Probability values ≤ 0.05 were considered statistically significant.

## 5. Conclusions

In conclusion, this study provides preliminary clinical evidence that once-weekly semaglutide administration results in significant weight and Hb_A1c_ reductions, while promoting favorable remodeling of atherogenic lipoprotein subfractions in T2DM. After 52 weeks of therapy, semaglutide produced significant decreases in LDL and non-HDL cholesterol, selectively reduced small, dense LDL subfractions, and elevated total HDL cholesterol with a shift toward larger, cardioprotective HDL particles. In contrast, sitagliptin demonstrated only modest glycemic and weight-lowering effects, with minimal effects on lipid profiles or inflammatory markers. These results emphasize the pleiotropic cardiometabolic benefits of GLP-1 RA beyond glycemic control and support the role of semaglutide in overall risk reduction. Future studies should explore the long-term cardiovascular outcomes associated with these lipoprotein changes and elucidate the underlying molecular mechanisms, particularly in diverse patient populations. Semaglutide has been identified as a potential therapeutic approach to treat multiple aspects of T2DM, including, but not limited to, optimizing glucose homeostasis, regulating body composition, and mitigating cardiovascular risk factors related to lipid metabolism.

## Figures and Tables

**Figure 1 ijms-26-05951-f001:**
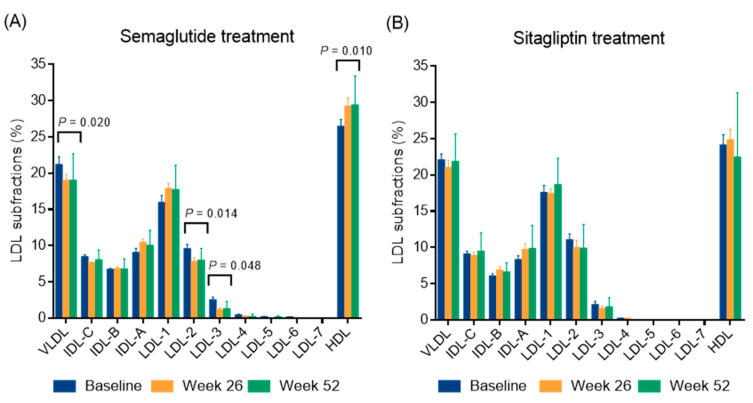
The distribution of lipoprotein subfractions in type 2 diabetic patients during a 52-week semaglutide (**A**) and sitagliptin treatment (**B**). The distribution of lipoprotein subfractions was analyzed using a Lipoprint gel electrophoresis system. Abbreviations: IDL, intermediate density lipoprotein; HDL, high-density lipoprotein; LDL, low-density lipoprotein; VLDL, very low-density lipoprotein.

**Figure 2 ijms-26-05951-f002:**
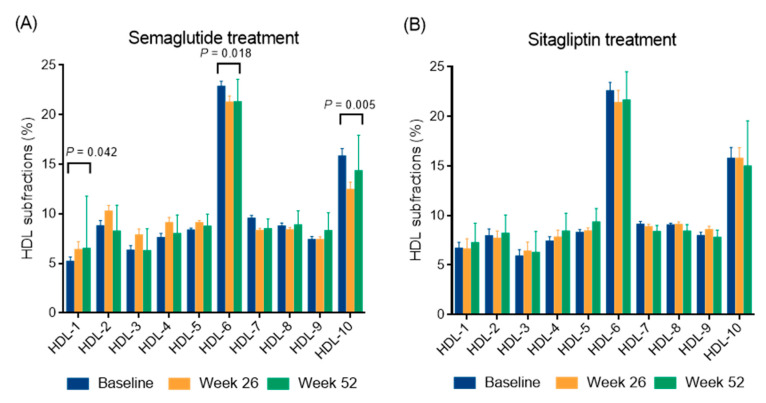
The distribution of high-density lipoprotein (HDL) subfractions in type 2 diabetic patients during a 52-week semaglutide (**A**) and sitagliptin treatment (**B**). The distribution of HDL subfractions was analyzed using a Lipoprint gel electrophoresis system.

**Figure 3 ijms-26-05951-f003:**
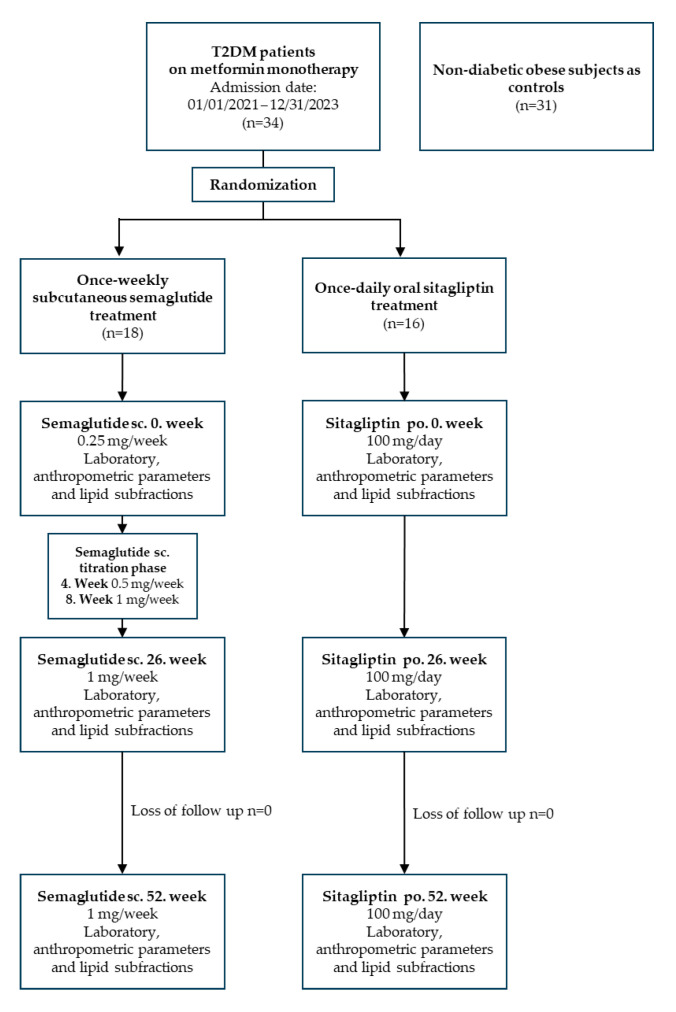
The flowchart of study design. Abbreviations: T2DM, type 2 diabetes mellitus; sc., subcutaneous; po., per os.

**Table 1 ijms-26-05951-t001:** The anthropometric and laboratory parameters of patients with T2DM before semaglutide and sitagliptin treatment and in the control group.

	Patients with T2DM		Controls
	Semaglutide Group (Before Treatment)	Sitagliptin Group (Before Treatment)	
Sex (male/female)	18 (12/6)	16 (7/9)	31 (10/21)
Age (year)	57.9 ± 10.6	59.4 ± 11.9	55.2 ± 5.0
BMI (kg/m^2^)	37.96 ± 10.64	31.26 ± 2.75	37.89 ± 9.64
Waist circumference (cm)	126.4 ± 21.8	126.6 ± 25.1	114.8 ± 21.0
Glucose (mmol/L)	8.1 (7.10–11.80) #	8.90 (7.60–10.80) §	5.35 (4.95–6.05)
Fructosamine (mmol/L)	322.39 ± 87.95 #	299.00 ± 68.80 §	239.35 ± 55.37
Hb_A1c_ (%)	8.1 ± 1.7 #	8.1 ± 1.3 §	6.0 ± 1.3
Insulin (mU/L)	19.4 (15.2–27.5)	26.2 (20.5–41.6) §	13.7 (8.9–19.8)
C-peptide (pmol/L)	1370 (1270–1800) #	1430 (1140–2730) §	826.5 (502–976)
Creatinine (µmol/L)	78.50 ± 15.07	71.47 ± 10.18	68.41 ± 15.21
eGFR (mL/min/1.73 m^2^)	87 (74–90)	90 (73–90)	90 (89–90)
hsCRP (mg/L)	2.3 (1.7–5.8) #	5.5 (2.9–13.8)	11.6 (3.0–18.0)
AST (U/L)	21 (16–25)	27 (21–35)	20.5 (17–24)
ALT (U/L)	24 (17–37)	34 (23–46)	25 (16–37)
GGT (U/L)	33 (24–49)	38 (22–87)	27 (18–39)
Triglyceride (mmol/L)	1.72 (1.50–3.11)	2.00 (1.30–3.40)	1.70 (1.0–2.28)
Total cholesterol (mmol/L)	5.49 ± 1.36	5.85 ± 2.24	5.14 ± 0.99
HDL cholesterol (mmol/L)	1.27 ± 0.30	1.34 ± 0.52	1.30 ± 0.32
Non-HDL cholesterol (mmol/L)	3.98 ± 0.95	4.09 ± 1.95	3.58 ± 1.29
LDL cholesterol (mmol/L)	3.13 ± 0.88	3.28 ± 1.51	3.28 ± 0.84
ApoA1 (g/L)	1.55 ± 0.24	1.62 ± 0.33	1.59 ± 0.27
ApoB (g/L)	1.03 ± 0.28	1.26 ± 0.50	1.04 ± 0.25
Large LDL {1–2} (mmol/L)	1.37 (1.14–1.48)	1.45 (1.17–2.25)	1.49 (1.18–1.62)
Small LDL {3–7} (mmol/L)	0.1 (0.05–0.22)	0.1 (0–0.21)	0.0 (0.0–0.09)
Mean LDL size (nm)	27.0 (26.5–27.1)	27.0 (26.6–27.3)	27.3 (27.0–27.4)
Large HDL {1–3} (mmol/L)	0.25 (0.17–0.28)	0.25 (0.16–0.39)	0.29 (0.22–0.36)
Intermediate HDL {4–7} (mmol/L)	0.58 (0.5–0.68)	0.56 (0.47–0.67)	0.62 (0.50–0.73)
Small HDL {8–10} (mmol/L)	0.41 (0.31–0.47)	0.39 (0.32–0.44)	0.39 (0.27–0.52)

Abbreviations: ApoA1, apolipoprotein AI; ApoB, apolipoprotein B100; BMI, body mass index; eGFR, estimated glomerular filtration rate; AST, aspartate transaminase, ALT, alanine transaminase; GGT, gamma-glutamyl transferase; Hb_A1c_, hemoglobin A1c; HDL, high-density lipoprotein; hsCRP, high-sensitivity C-reactive protein; LDL, low-density lipoprotein. The comparison of data was made with Kruskal–Wallis ANOVA, # marks *p* < 0.05 between semaglutide and the control group; § marks *p* < 0.05 between sitagliptin and the control group.

**Table 2 ijms-26-05951-t002:** The change in laboratory parameters in patients with T2DM after a 52-week semaglutide treatment.

	Baseline	Week 26	Week 52
BMI (kg/m^2^)	37.96 ± 10.64	35.28 ± 9.43 †	34.88 ± 10.22 §
Waist circumference (cm)	126.4 ± 21.8	119.7 ± 21.6 †	115.5 ± 20.9 §
Glucose (mmol/L)	8.1 (7.1–11.8)	7.5 (5.0–8.6) †	7.7 (5.3–10.4) §
Fructosamine (mmol/L)	322.4 ± 87.9	260.4 ± 39.2 †	251.5 ± 37.7 §
Hb_A1c_ (%)	8.08 ± 1.65	6.86 ± 1.12 †	6.57 ± 0.95 §
Insulin (mU/L)	19.4 (15.2–27.5)	21.6 (12.3–28.1)	19.3 (9.1–30.2)
C-peptide (pmol/L)	1370 (1270–1800)	1340 (733–2110)	1240 (1040–1610)
Creatinine (µmol/L)	78.50 ± 15.07	77.94 ± 30.64	71.56 ± 16.67
eGFR (mL/min/1.73 m^2^)	86.5 (74–90)	90 (78–90)	90 (84–90)
hsCRP (mg/L)	2.3 (1.7–5.8)	2.30 (1.30–6.26)	2.10 (1.10–2.40)
AST (U/L)	21 (17–25)	20.5 (14–28)	17 (15–26)
ALT (U/L)	27 (21–37)	26 (18–32)	21 (16–31)
GGT (U/L)	34 (27–56)	33 (23–45)	35 (22–50)
Triglyceride (mmol/L)	1.72 (1.50–3.11)	1.575 (1.00–2.17)	1.585 (1.0–2.50)
Total cholesterol (mmol/L)	5.49 ± 1.36	4.83 ± 1.31	4.79 ± 1.00
HDL cholesterol (mmol/L)	1.27 ± 0.30	1.40 ± 0.37 †	1.43 ± 0.38 §
Non-HDL cholesterol (mmol/L)	3.98 ± 0.95	3.34 ± 1.08	3.35 ± 0.82 §
LDL cholesterol (mmol/L)	3.13 ± 0.88	2.85 ± 1.06	2.72 ± 0.84 §
Mean LDL size (nm)	27.0 (26.5–27.1)	27.2 (27.1–27.3) †	27.1 (26.9–27.3) §
Large LDL {1–2} (mmol/L)	1.37 (1.14–1.48)	1.17 (0.9–1.7) #	1.28 (1–1.5) §
Small LDL {3–7} (mmol/L)	0.1 (0.05–0.2)	0.02 (0–0.1) #	0.06 (0–0.1) §
Large HDL {1–3} (mmol/L)	0.25 (0.2–0.3)	0.27 (0.2–0.4) #	0.24 (0.2–0.3)
Intermediate HDL {4–7} (mmol/L)	0.58 (0.5–0.7)	0.66 (0.6–0.8) #	0.66 (0.6–0.7) §
Small HDL {8–10} (mmol/L)	0.41 (0.3–0.5)	0.41 (0.3–0.5)	0.41 (0.3–0.5)
ApoA1 (g/L)	1.55 ± 0.24	1.49 ± 0.23	1.54 ± 0.27
ApoB (g/L)	1.03 ± 0.28	0.96 ± 0.31	0.99 ± 0.25

Abbreviations: ApoA1, apolipoprotein AI; ApoB, apolipoprotein B100; BMI, body mass index; eGFR, estimated glomerular filtration rate; AST, aspartate transaminase, ALT, alanine transaminase; GGT, gamma-glutamyl transferase; Hb_A1c_, hemoglobin A1c; HDL, high-density lipoprotein; hsCRP, high-sensitivity C-reactive protein; LDL, low-density lipoprotein. To compare the data, we used repeated measures ANOVA and applied Bonferroni correction. † marks *p* < 0.05 between baseline and week 26; § marks *p* < 0.05 between baseline and week 52, # marks *p* <0.05 between week 26 and week 52.

**Table 3 ijms-26-05951-t003:** The change in laboratory parameters in patients with T2DM after a 52-week sitagliptin treatment.

	Baseline	Week 26	Week 52
BMI (kg/m^2^)	31.26 ± 2.75	31.10 ± 2.98	30.911 ± 2.9 §
Waist circumference (cm)	126.60 ± 25.08	125.25 ± 29.18	124.25 ± 29.58
Glucose (mmol/L)	8.90 (7.60–10.80)	7.50 (5–7.9) †	8.00 (5.7–9.3)
Fructosamine (mmol/L)	299.0 ± 68.8	262.6 ± 51.2	253.4 ± 28.7 §
Hb_A1c_ (%)	8.13 ± 1.29	6.91 ± 0.95 †	7.11 ± 1.07 §
Insulin (mU/L)	26.2 (20.5–41.6)	26.0 (18.7–40.2)	21.95 (12.5–26.5)
C-peptide (pmol/L)	1430 (1140–2730)	1870 (966–2020)	1110 (904.5–1675)
Creatinine (µmol/L)	71.47 ± 10.18	70.571 ± 15.301	71.20 ± 14.26
eGFR (ml/min/1.73 m^2^)	90 (73–90)	90 (85–90)	90 (85–90)
hsCRP (mg/L)	5.5 (2.9–13.8)	4.7 (2.3–11.7)	8.7 (2.4–14.7)
AST (U/L)	27 (21–35)	23 (18–32)	28.5 (16–36)
ALT (U/L)	34 (23–46)	29.5 (21–43)	36 (25–46)
GGT (U/L)	38 (22–87)	39 (20–52)	39 (25–44)
Triglyceride (mmol/L)	2.00 (1.30–3.40)	2.08 (1.55–2.9)	1.850 (1.3–3.1)
Total cholesterol (mmol/L)	5.85 ± 2.24	5.264 ± 1.32	5.31 ± 1.57
HDL cholesterol (mmol/L)	1.34 ± 0.52	1.303 ± 0.44	1.338 ± 0.34
Non-HDL cholesterol (mmol/L)	4.09 ± 1.95	3.962 ± 1.34	4.00 ± 1.52
LDL cholesterol (mmol/L)	3.28 ± 1.51	3.092 ± 1.08	3.209 ± 1.2
Mean LDL size (nm)	27.0 (26.6–27.3)	27.1 (26.7–27.2)	27.0 (26.7–27.3)
Large LDL {1–2} (mmol/L)	1.45 (1.17–2.3)	1.47 (1.2–1.6)	1.4 (1.3–1.9)
Small LDL {3–7} (mmol/L)	0.1 (0–0.2)	0.07 (0–0.2)	0.09 (0–0.1)
Large HDL {1–3} (mmol/L)	0.25 (0.2–0.4)	0.24 (0.1–0.4)	0.28 (0.2–0.4)
Intermediate HDL {4–7} (mmol/L)	0.56 (0.5–0.7)	0.54 (0.4–0.7)	0.59 (0.58–0.8)
Small HDL {8–10} (mmol/L)	0.39 (0.3–0.4)	0.37 (0.3–0.5)	0.39 (0.35–0.43)
ApoA1 (g/L)	1.62 ± 0.33	1.528 ± 0.39 †	1.484 ± 0.34 §
ApoB (g/L)	1.26 ± 0.50	1.10 ± 0.4	1.197 ± 0.41

Abbreviations: ApoA1, apolipoprotein AI; ApoB, apolipoprotein B100; BMI, body mass index; eGFR, estimated glomerular filtration rate; AST, aspartate transaminase, ALT, alanine transaminase; GGT, gamma-glutamyl transferase; Hb_A1c_, hemoglobin A1c; HDL, high-density lipoprotein; hsCRP, high-sensitivity C-reactive protein; LDL, low-density lipoprotein. To compare the data, we used repeated measures ANOVA test and applied Bonferroni correction. † marks *p* < 0.05 between baseline and week 26, § marks *p* < 0.05 between baseline and week 52.

**Table 4 ijms-26-05951-t004:** Changes in lipid subfractions induced by semaglutide treatment in relation to changes in BMI.

	ΔBMI *	*p*-Value
Δ Large LDL {1–2} (mmol/L)	0.01	0.680
Δ Small LDL {3–7} (mmol/L)	−0.03	0.390
Δ Mean LDL size (nm)	0.23	0.534
Δ Large HDL {1–3} (mmol/L)	0.23	0.534
Δ Intermediate HDL {4–7} (mmol/L)	0.01	0.310
Δ Small HDL {8–10} (mmol/L)	0.01	0.310

Abbreviations: HDL, high-density lipoprotein; LDL, low-density lipoprotein. We used multivariate linear regression analysis to assess the association between changes in LDL and HDL subfractions and improvement in BMI. * Changes in BMI values were adjusted for age, gender, and changes in HbA1c values.

## Data Availability

All data generated or analyzed during this study are included in this published article. All data generated or analyzed during this study are available from the corresponding author upon reasonable request.

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
