# Peer review of "Semaglutide Improves Lipid Subfraction Profiles in Type 2 Diabetes: Insights from a One-Year Follow-Up Study"

_ijms, 2025, doi:10.3390/ijms26135951_

Round 1

Reviewer 1 Report

Comments and Suggestions for Authors

The manuscript entitled "Semaglutide Improves Lipid Subfraction Profiles in Type 2 Diabetes: Insights from a One-Year Follow-Up Study" presents a well-structured and clinically relevant investigation into the effects of semaglutide and sitagliptin on lipoprotein subfractions in patients with type 2 diabetes mellitus (T2DM). The study is timely and contributes to the growing body of literature on the pleiotropic effects of GLP-1 receptor agonists, particularly in the context of cardiovascular risk modulation.

However, I would like to raise several points for clarification and potential improvement:

Point 1. The authors reference Clinical Endocrinology (2014) 81, 370–377 for the stratification of LDL and HDL particle sizes. It would enhance the clarity and reproducibility of the study if the authors could:

  • Include visual representations (e.g., electrophoresis images or schematic diagrams) in the supplementary materials to illustrate the analytical process.
  • Explicitly describe in the main text the classification criteria used to define large, intermediate, and small HDL subfractions (e.g., HDL1–3 as large, HDL4–7 as intermediate, HDL8–10 as small), as cited in the referenced study.

Point 2 . The manuscript includes a non-diabetic obese control group; however, this group does not appear to be incorporated into the main comparative analyses. The authors should clarify the intended role of this group. If it was not used in statistical comparisons or mechanistic interpretation, its inclusion may be reconsidered or better justified.

Point 3. While the current findings are valuable, the study could be further strengthened by incorporating additional analyses of lipoprotein composition and function. For example:

  • Proteomic and lipidomic profiling of LDL and HDL particles.
  • Quantification of phospholipid and cholesterol ester content.
  • Functional assays, particularly HDL-mediated cholesterol efflux capacity in macrophages, which would provide mechanistic insight into the observed changes.

This study represents a meaningful step toward understanding the cardiometabolic benefits of semaglutide in T2DM. Addressing the points above would enhance the scientific rigor and translational relevance of the manuscript. I encourage the authors to consider these suggestions in their revision.

Author Response

Ms. Liane Zhu

Assigned Editor

and

Dr. Edit Szabó

Special Issue Editor

Special issue: Type 2 Diabetes: Molecular Pathophysiology and Treatment

Dear Editors,

we have received the editorial response and the reviewers’ comments and questions regarding our revised manuscript titled “Semaglutide Improves Lipid Subfraction Profiles in Type 2 Diabetes: Insights from a One-Year Follow-Up Study" (ijms-3680168). Thank you for the opportunity to submit our revised manuscript to your journal.

Please find below our point by point answers to the reviewers’ questions. Changes in the revised manuscript are marked on red. We also submit the final corrected version of the manuscript without markups.

Response to Reviewer 1.

The manuscript entitled "Semaglutide Improves Lipid Subfraction Profiles in Type 2 Diabetes: Insights from a One-Year Follow-Up Study" presents a well-structured and clinically relevant investigation into the effects of semaglutide and sitagliptin on lipoprotein subfractions in patients with type 2 diabetes mellitus (T2DM). The study is timely and contributes to the growing body of literature on the pleiotropic effects of GLP-1 receptor agonists, particularly in the context of cardiovascular risk modulation.

Thank you for the thoughtful review and the positive comments on our manuscript.

Responses to comments:

However, I would like to raise several points for clarification and potential improvement:

Point 1. The authors reference Clinical Endocrinology (2014) 81, 370–377 for the stratification of LDL and HDL particle sizes. It would enhance the clarity and reproducibility of the study if the authors could:

  • Include visual representations (e.g., electrophoresis images or schematic diagrams) in the supplementary materials to illustrate the analytical process.
  • Explicitly describe in the main text the classification criteria used to define large, intermediate, and small HDL subfractions (e.g., HDL1–3 as large, HDL4–7 as intermediate, HDL8–10 as small), as cited in the referenced study.

Response:

  • In accordance with the Reviewer's request, in the supplementary materials (Supplementary Image S1), we provided a visual depiction of the electrophoresis images of the LDL and HDL subfractions of each patient to illustrate the analytical process. The reference was inserted into the manuscript: „We visually depicted the electrophoresis images of the LDL and HDL subfractions of each patient in the supplementary materials (Supplementary Figure S1 and S2) to illustrate the analytical process.” (line 463-465).
  • We added the following sentence to the "Lipoprotein subfraction analyses" section: „After electrophoresis, lipoprotein fractions (bands) were identified by their mobility (Rf) using very low-density lipoprotein (VLDL) as the starting reference point (Rf 0.0) and high-density lipoprotein (HDL) as the ending reference point (Rf 1.0). The LDL subfractions were distributed between these two points, from LDL1 to LDL7 (Rf 0.32, 0.38, 0.45, 0.51, 0.56, 0.6, and 0.64, respectively). The LDL1 and LDL2 bands corresponded to large, buoyant LDL subclasses, while the LDL3 to LDL7 bands corresponded to small, dense LDL subclasses. The percentages of the area under the curve (AUC%) were calculated using Lipoware computer software (Quantimetrix Corp.).” ((line 440-447)
  • We also added the following sentence to the "Lipoprotein subfraction analyses" section: “The stained HDL subfractions (bands) were identified by their mobility after electrophoresis. The LDL/VLDL band served as the initial reference point (Rf 0.0), while albumin served as the final reference point (Rf 1.0). AUC% were calculated using the Lipoware computer software (Quantimetrix Corp.).” (line 455-459)

Point 2 . The manuscript includes a non-diabetic obese control group; however, this group does not appear to be incorporated into the main comparative analyses. The authors should clarify the intended role of this group. If it was not used in statistical comparisons or mechanistic interpretation, its inclusion may be reconsidered or better justified.

Response:

  • In this study, a cohort of 31 obese but non-diabetic individuals, matched for age and gender, served as the control group. Their inclusion enabled baseline metabolic and lipoprotein characteristics to be assessed in the absence of type 2 diabetes mellitus (T2DM), facilitating comparison with T2DM patients prior to the initiation of semaglutide or sitagliptin treatment. Importantly, these controls were free from any medication, eliminating potential pharmacological confounders. A comparison of baseline clinical and laboratory parameters showed that, although there were no significant differences in age, BMI, waist circumference, renal function, liver enzymes or lipid profile between diabetic patients and the control group, key glycaemic indices — including fasting glucose, fructosamine, HbA1c and C-peptide — were significantly higher in both treatment groups of diabetic patients than in the control group (p < 0.05). Additionally, although LDL and HDL subfraction levels did not differ significantly between groups, the T2DM groups exhibited a trend towards smaller LDL particle size and lower concentrations of large HDL subfractions compared to the control group. These findings emphasise the metabolic differences associated with T2DM beyond obesity alone, and demonstrate the value of the control group in distinguishing diabetes-related changes from those caused solely by excess adiposity.
  • We added the following sentence to the „Methods” section: „Non-diabetic, obese control subjects were included to distinguish metabolic alterations related to type 2 diabetes from those associated with obesity alone.” (line 391-393)

Point 3. While the current findings are valuable, the study could be further strengthened by incorporating additional analyses of lipoprotein composition and function. For example:

  • Proteomic and lipidomic profiling of LDL and HDL particles.
  • Quantification of phospholipid and cholesterol ester content.
  • Functional assays, particularly HDL-mediated cholesterol efflux capacity in macrophages, which would provide mechanistic insight into the observed changes.

Response:

  • Thank you very much for the Reviewer's comment. Evaluation of proteomic and lipidomic profiling of LDL and HDL particles, quantification of phospholipid and cholesterol ester content, or functional assays were not included in this study. These methods are not widely available and are associated with extremely high costs. Moreover, they were not covered by the ethical approval for this study. However, investigation of the above-mentioned parameters could be the focus of future research.. Therefore, we added the following sentences to the „Limitations”: „Another limitation of this study is the absence of a detailed analysis of lipoprotein particle composition and functional properties. While standard lipid parameters and subfraction distributions were examined, incorporating advanced proteomic and lipidomic profiling of LDL and HDL particles could have provided deeper insight into qualitative differences between groups. Additionally, quantifying phospholipid and cholesterol ester content and conducting functional studies, such as assessing HDL-mediated cholesterol efflux capacity in macrophages, would have improved our mechanistic understanding of the observed metabolic changes. (line 366-374)

This study represents a meaningful step toward understanding the cardiometabolic benefits of semaglutide in T2DM. Addressing the points above would enhance the scientific rigor and translational relevance of the manuscript. I encourage the authors to consider these suggestions in their revision.

Thank you again for your thorough review of our manuscript and your valuable feedback!

Response to Reviewer 2.

In this manuscript, the authors provide an insight on the topic titled "Semaglutide Improves Lipid Subfraction Profiles in Type 2 Diabetes: Insights from a One-Year Follow-Up Study". While this area of weight management and their possible benefits is emerging yet well researched, the authors should address the following concerns.

We would like to thank you for your valuable comments that improve our manuscript.

Responses to comments:

1) On page 2, line 10, potential readers could benefit if the authors define what PREVEND study is, and how it was used in this study.

Response:

  • The PREVEND study is a large, prospective, population-based cohort investigating the relationship between lipoprotein subfractions, particularly triglyceride-rich lipoproteins and LDL particle characteristics, and the development of type 2 diabetes in the general population.
  • In our study, we used insights from the PREVEND study to interpret baseline and treatment-induced changes in lipoprotein subfractions in patients with type 2 diabetes treated with semaglutide or sitagliptin. This allowed us to explore the potential implications of these changes for diabetes-related dyslipidemia and cardiometabolic risk.
  • We added the following sentence to the „Introduction” section: „Emerging evidence from the large, prospective, population-based PREVEND study suggests that distinct patterns of association exist between obesity, triglyceride levels, LDL particle characteristics, and the risk of developing T2DM. Specifically, elevated levels of TRLs are associated with an increased risk of T2DM. In contrast, larger LDL particle size and greater LDL particle diameter appear to be associated with a decreased risk of diabetes, potentially providing protection against the development of the disease.” (line 56-62)

2) The whole manuscript is rife with many acronyms which could potentially serve as a source of distraction. It would be helpful if the author minimizes the use of these acronyms throughout the manuscript. The authors should consider revising this.

Response:

  • The Reviewer is absolutely right; we have reduced the number of abbreviations and standardized them. (Abbreviation)

3) Under discussion, line 11, It would be helpful to readers if the authors explain what SUSTAIN-6 trial is, how it was conducted, and how it was used in this study.

Response:

  • In accordance with the Reviewer's request, we rephrased the sentences in the Discussion: „The SUSTAIN-6 trial, a multicenter, randomized, placebo-controlled study, demonstrated that once-weekly subcutaneous administration of 1 mg semaglutide in patients with type 2 diabetes mellitus resulted in a significant 1.5% reduction in HbA1c, an average weight loss of approximately 4.5 kg, and a 26% decrease in major adverse cardiovascular events over 104 weeks. In the present study, we applied the findings from SUSTAIN-6 to assess the effects of semaglutide on anthropometric, glycemic, and lipid parameters over 52 weeks in obese patients with type 2 diabetes, observing consistent reductions in HbA1c and body weight, along with favorable changes in lipoprotein subfractions and lipid profiles, thereby corroborating and extending the evidence from large randomized clinical trials. “ (line 223-232)

4) While number of participants in this study is very limited, it would be helpful if the authors expand/increase the number of participants in this study to include subjects from different race as well.

Response:

  • The Reviewer is absolutely right; we acknowledge that the limited number of participants and the homogeneity of the study population represent important limitations. Due to resource and time constraints, expanding the sample size and including subjects from diverse racial backgrounds was not feasible within the scope of the current study. Indeed, the Hungarian population is highly homogeneous in terms of racial background. However, we fully agree that future research should aim to include a larger and more ethnically diverse cohort to enhance the generalizability of the findings. We have addressed this point in the revised manuscript’s limitations section and highlighted the need for further studies in more diverse populations.
  • We rephrase the following sentence in the „Limitation”: „The single-centre design, limited sample size, and lack of racial diversity restrict the generalisability of the results, highlighting the need for further studies in more diverse populations.” (line 359-361)

5) It would also be helpful to know if the diabetic subjects in this study were on some form of medications and how the results obtained were impacted by those medications.

Response:

  • Thank you very much for the Reviewer's comment. In this study, all patients were on metformin monotherapy at baseline. Additionally, all participants received statin therapy, and the majority were also treated with either an ACE inhibitor, an angiotensin receptor blocker, or a beta-blocker therapy. We acknowledge that these concomitant medications may have influenced the study outcomes, and we have now included this information in the revised manuscript to provide a clearer context for interpreting the results.
  • We added the following sentence to the „Methods”: „All diabetic participants were on metformin monotherapy, received statin therapy, and the majority were also treated with an ACE inhibitor, angiotensin receptor blocker, or beta-blocker” (line 387-389)

In summary, while this area of research and its benefits continue to grow, this manuscript could potentially benefit its target audience if the authors address the above concerns.

Thank you again for your thorough review of our manuscript and your valuable feedback!

Reviewer 2 Report

Comments and Suggestions for Authors

In this manuscript, the authors provide an insight on the topic titled "Semaglutide Improves Lipid Subfraction Profiles in Type 2 Diabetes: Insights from a One-Year Follow-Up Study". While this area of weight management and their possible benefits is emerging yet well researched, the authors should address the following concerns.

1) On page 2, line 10, potential readers could benefit if the authors define what PREVEND study is, and how it was used in this study.

2) The whole manuscript is rife with many acronyms which could potentially serve as a source of distraction. It would be helpful if the author minimizes the use of these acronyms throughout the manuscript. The authors should consider revising this.

3) Under discussion, line 11, It would be helpful to readers if the authors explain what SUSTAIN-6 trial is, how it was conducted, and how it was used in this study.

4) While number of participants in this study is very limited, it would be helpful if the authors expand/increase the number of participants in this study to include subjects from different race as well.

5) It would also be helpful to know if the diabetic subjects in this study were on some form of medications and how the results obtained were impacted by those medications.

In summary, while this area of research and its benefits continue to grow, this manuscript could potentially benefit its target audience if the authors address the above concerns.

Author Response

(The authors gave the same response as above.)
